# A Bibliometric Analysis of High-Intensity Interval Training in Cardiac Rehabilitation

**DOI:** 10.3390/ijerph192113745

**Published:** 2022-10-22

**Authors:** Haitao Liu, Feiyue Liu, Haoyuan Ji, Zuanqin Dai, Wenxiu Han

**Affiliations:** 1College of Physical Education, Henan University, Kaifeng 475001, China; 2Research Center of Sports Reform and Development, Henan University, Kaifeng 475001, China; 3Institute of Physical Fitness and Health, Henan University, Kaifeng 475001, China

**Keywords:** high-intensity interval training, cardiac rehabilitation, bibliometrics, CiteSpace, cluster analysis, co-citation analysis

## Abstract

As global quality of life has improved, the risk factors for cardiovascular diseases have gradually increased in prevalence. People have consequently sought to improve their health through physical exercise. High-intensity interval training (HIIT) is a cardiac rehabilitation (CR) tool that has been of great interest for several years. However, its feasibility and safety remain controversial. This study aimed to explore hot research topics and new directions regarding the role of HIIT in CR and to describe the dynamic development of the field. We used the Web of Science Core Collection database to develop visualizations using CiteSpace software (v.6.1.R2). The number of articles published, institutional collaboration networks, author partnerships, and keyword co-occurrence and clustering were used to analyze the impact of HIIT on CR. Our results showed that Norway, Canada, and the United States were the most prominent contributors to this field. Articles by Nigam, A and Juneau, M had the highest number of citations. The Norwegian University of Science and Technology had performed the most in-depth research in this area. The *European Journal of Preventive Cardiology* had published the most articles. The United States had the highest number of publishing journals. Relevant issues focused on coronary artery disease, exercise capacity, heart failure, cardiorespiratory fitness, and physical activity. HIIT in heart transplantation may be at the forefront of research in this field and future studies should focus on this topic. HIIT-based CR can therefore improve the exercise capacity and quality of life of cardiovascular patients and improve patient compliance in a safe manner.

## 1. Introduction

Heart disease (HD), as a relatively common disease of the circulatory system, is a general term [1] that refers to cardiac injury or abnormal heart function. Heart disease can be congenital (CHD) [2] or acquired. Common symptoms of HD include chest tightness, chest pain, palpitations, shortness of breath, fatigue, dizziness, and sweating, with chest pain being the most typical symptom [3]. Heart-disease-associated morbidity and mortality rates have increased in recent years, affecting 30–40% of patients over 60 years of age [4]. This is mainly due to the patients ignoring their health and external environment for a long period of time. This leads to high levels of circulating triglycerides [5], which can contribute to atherosclerosis—the most common cause of heart disease. Heart disease can lead to sudden cardiac death, representing a serious threat to human health [6].

Cardiac rehabilitation (CR) aims to improve the daily living and quality of life of patients with HD through a comprehensive rehabilitation strategy to prevent the recurrence of cardiovascular disease [7]; it includes rehabilitation assessment, regular medication, exercise therapy, diet therapy, behavioral therapy, and health education [8]. Cardiac rehabilitation is generally aimed towards patients with stable angina, atherosclerosis, heart transplants [9], and/or cardiac infarctions [10]. Exercise therapy is a very important part of CR. Physical function and the ability to perform daily work and life activities can be improved by exercise [11]. Exercise also improves vascular endothelial function [12] and resistance to disease, increases the excitability of the nervous system, enhances the activity of cardiomyocytes, and reduces cardiovascular risk factors and mortality [13].

Moderate intensity continuous training (MICT) is the preferred exercise modality for CR because of its high safety and effectiveness [14]. However, high-intensity interval training (HIIT) has recently been found to be more effective than MICT in the CR of patients with cardiovascular disease [15]. High-intensity interval training is a training method in which the practitioner exercises with extreme intensity (i.e., equal to or greater than the anaerobic threshold or maximum lactate steady-state load intensity) for a few seconds to a few minutes with sustained effort interrupted by a short, inadequate rest interval so that the body is in an incomplete recovery state [16]. High-intensity interval training has been shown to be safe and is used in the rehabilitation of patients with chronic diseases, as well as for physical fitness by the general population [17].

Bibliometric analysis is a key process for further understanding of a field through functions such as clustering, which can help to determine research topics and sort through large amounts of literature data [18]. CiteSpace is a citation visualization analysis software platform developed in the context of scientometrics and data visualization [19] that analyzes literature citations in specific regions and presents the structure, laws, and distribution of scientific knowledge through visual graphs [20]. CiteSpace is now widely used to study disciplinary dynamics and hotspots and can clarify the research status of a field by analyzing the number of articles published, institutional collaboration networks, author partnerships, and keyword co-occurrence and clustering. Therefore, this study aimed to use CiteSpace to explore hot research topics and new directions regarding the role of HIIT in CR, and to describe the dynamic development of the field.

## 2. Data Collection and Research Method

### 2.1. Data Collection

Data from the relevant literature used in this study were extracted from the Web of Science (WOS) database, with a timeline from 1 January 2000 to 15 August 2022. The literature inclusion criteria were as follows: literature that was closely related to the study topic, and literature investigating both HIIT and CR. The literature exclusion criteria were as follows: studies investigating only HIIT or CR, or non-heart-related disease, and literature examining other forms of exercise. The first step was to run the topic search “TS = ‘high-intensity interval training or high intensity interval training or HIIT and cardiac rehabilitation’ and TS = ‘high-intensity interval training or high intensity interval training or HIIT and heart’” as the topic, where 2151 items were retrieved, and 181 articles were retained after manual screening. The second step was to search the author keywords (AK) “AK = high-intensity interval training or high intensity interval training or HIIT and coronary artery disease or myocardial infarction or percutaneous transluminal coronary or heart failure or heart transplant”, where 51 papers were retrieved, and 13 were retained after manual screening. The third step was to run the author keyword search “AK = high-intensity interval training or high intensity interval training or HIIT and heart”, which retrieved 76 papers, and 7 were retained after manual screening. A total of 201 relevant items were ultimately obtained on the same day (Figure 1). These retrieved studies were exported in plain text format with full records and cited references and imported into CiteSpace software. The formatted data files in the “output” folder were copied and pasted into the “data” folder for subsequent data analysis.

### 2.2. Research Methods

CiteSpace visualization software based on the JAVA platform was used to map scientific knowledge. Co-occurrence and clustering analyses were conducted from 1 January 2000 to 15 August 2022 in the WOS database for “high-intensity interval training”, “cardiac rehabilitation”, and related high-frequency terms, research institutions, and countries. Trend graphs of publication volume, authors, geographic distribution, and keywords were drawn to highlight key nodes and research hotspots, thereby visualizing the evolution of HIIT in CR [20]. CiteSpace captured research trends regarding HIIT in CR, explored the key paths and turning points in the evolution of this field, and formed a series of visualization maps to explore the frontiers of the field’s development [21].

## 3. Results

### 3.1. Time Distribution

An analytical search of the WOS database yielded 201 publications by foreign scholars from 2004 to 2022. The research on HIIT in the field of cardiac rehabilitation can be seen in Figure 2 from 2004 to 2011 as a steady phase and from 2012 to 2016 as a slow growth phase. However, from 2017 onwards there was a period of rapid year-over-year growth, and a peak was reached in 2021. 

### 3.2. Journals Distribution

The journals were ranked by the number of publications. The top nine journals that published these articles were also identified, with citation frequency, impact factor (IF), and other journal information shown in Table 1. The nine journals with the highest interest in the impact of HIIT on CR published 62 articles on the topic, accounting for 30.8% of all of the included studies. The *European Journal of Preventive Cardiology* was ranked first with 15 articles, followed by the *International Journal of Cardiology* and the *Journal of Cardiopulmonary Rehabilitation and Prevention*, with 9 articles each. From Table 1, we can see that the journal Circulation from the United States ranked first in IF (39.918), CiteScore (40.5), and citation frequency (505).

### 3.3. Highly Productive Paper Authors

The time range was set to 2000–2022, with a time slice of 1 year and a g-index parameter k set to 25 in the selection criteria (Figure 3). There were 332 nodes with 836 connections and a network density of 0.0152. As can be seen from Table 2, Authors with 11 publications included Gullestad, L and Nytroen, K from Norway, and Juneau, M and Nigam, A from Canada, tied for first place. Among the top 10 authors in terms of number of publications, 4 was from Norway and 3 was from Canada. Among them, Juneau, M and Nigam, A from Canada ranked first in terms of total citations (426).

### 3.4. Analysis of Cited Articles

Citation frequency is an important indicator of the influence and quality of the scientific work of countries, institutions, and individuals. Table 3 shows that the most frequently cited article was from *Circulation*, with 269 total citations. This paper was published by Rognmo, O in 2012. Of the top 10 most frequently cited articles, 2 articles were from Circulation. Among the top 10, 2 articles each were published in 2012, 2015, and 2017.

### 3.5. Country Distribution

The countries with more than 20 articles were Norway (33), Canada (30), the United States (29), and China (21), accounting for 16.4%, 14.9%, 14.4%, and 10.4% of the total number of articles published, respectively (Table 4). According to the literature, scholars from Norway, Canada, and the United States were the first to publish on the topic, and their research productivity has been at a stable level in recent years. Other countries with significant numbers of publications included Australia (19), Germany (13), Brazil (13), the United Kingdom (12), Spain (12), and Switzerland (11) (Figure 4).

### 3.6. Institutional Distribution

The four institutions with 10 or more publications in this field from 2000 to 2022 were the Norwegian University of Science and Technology (14 articles), University of Oslo (13 articles), University of Queensland (12 articles), and University of Montreal (10 articles) (Table 5). The top 10 institutions were mainly from Norway, Canada, Australia, the United States, and Germany; of these, 4 institutions were from Norway and Canada, 2 institutions were from Australia, 1 institution was from Germany, and 1 was from the United States.

### 3.7. Keyword Analysis

#### 3.7.1. Keyword Co-Occurrence Analysis

In the functional parameter area of CiteSpace, “Keyword” was selected as the network node, the time slice was 1 year, and the threshold value was set to “Top N = 50%” to visualize the keyword co-occurrence of HIIT in CR; the results are shown in Figure 5. The number of nodes in Figure 5 is 283, the network density is 0.0463, and there are 1849 lines between the nodes. The number of lines indicates the increased co-occurrence of two keywords, the color of the lines indicates the year of occurrence of the keywords, and the thickness of the lines indicates the closeness of the connection between the keywords. In addition to high-frequency core keywords such as “coronary artery disease”, “exercise capacity”, “heart failure”, “cardiorespiratory fitness”, “quality of life”, “physical activity”, and “myocardial infarction” are shown in red in Figure 5, indicating a recent surge in interest. As can be seen in Table 6, many of the studies were performed to improve the physiological function of the human body through exercise interventions for patients with different cardiovascular diseases—including coronary artery disease and heart failure—and to identify the optimal intensity of HIIT.

#### 3.7.2. Keyword Clustering Analysis

Based on the keyword co-occurrence analysis, in order to better analyze the hot topics and trends of HIIT in the field of CR, a keyword timeline clustering analysis was performed (Figure 5); keyword clustering time-zone mapping was run using CiteSpace (Figure 6), and the specific time-zone mapping information is summarized in Table 7. Time-zone mapping yielded nine sets of related clustering labels; the smaller the number of labels, the more keywords were included in the clusters. The cluster module value (Q value) of the map was 0.3781, indicating a significant cluster structure. The average profile value (S value) of the clusters was 0.7187, which is a measure of homogeneity; the larger the S value, the higher the homogeneity. This value suggests that the clusters were reasonably homogeneous. Relevant keywords for research on the effects of HIIT on CR first appeared in 2004, and a second wave of relevant keywords appeared in 2010.

Based on the previous keyword co-occurrence mapping and clustering mapping analysis, CiteSpace’s emergent word detection function was further used to identify emergent words, emergent intensity, and emergent time to describe the use of a keyword over a specific time period (Figure 7). An emergent word refers to keywords that have changed significantly over a short period of time, and the red area indicates the length of time that the keyword has lasted from the year in which its sudden appearance began [22]. Emergent word analysis enabled us to observe a shift in HIIT research hotspots in CR in recent years. The key term “continuous moderate exercise” had the highest emergent intensity (3.08).

## 4. Discussion

Cardiac rehabilitation is designed to improve the function and structure of the heart in patients with heart disease and to optimize the patient’s physical and mental state [23]. Long-term HIIT intervention produces good results, increasing the patients’ ability to care for themselves and leading to an improved quality of life. CiteSpace provides analysis by visual means to present the structure, pattern, and distribution of scientific knowledge and summarize the relationships between authors, countries, journals, and institutions through visual mapping, which is now widely used to study disciplinary dynamics and hot spots.

Trends in the impact of HIIT on CR and the level of research interest in this area can be reflected by the number of papers published over a given time period. Our data show that the benefits of HIIT are being increasingly reported, particularly regarding its role and safety in CR. More relevant research findings are bound to emerge, as shown by the trendline of the index.

Among the top 10 journals with the highest number of publications, *Circulation* from the USA ranked first in IF, CiteScore, and citation frequency. The category of the journal is medicine, its subcategory is cardiology and cardiovascular medicine, and its rank in this category is 1/336. This indicates that the academic papers published in this journal are of high quality and have a strong influence. In terms of countries of publication, four of the top nine journals were from the USA, indicating that the USA has invested more and attached more importance to this area of study.

Collaboration mapping allows researchers to describe the contributions of individual scholars to the field and their collaborative relationships with one another, as well as to visually identify the more prolific authors who have published more papers [24]. Among them, Juneau, M and Nigam, A from Canada ranked first in terms of total citations, indicating that their scholarly work has been recognized by numerous scholars. Juneau, M and Nigam, A focused on the effects of HIIT intervention in patients with heart disease—particularly those with heart failure and reduced ejection fraction [25,26,27]. A comparison or combination of different forms of exercise was also performed to find the most suitable form of exercise for cardiac patients [28,29,30]. A comparison of HIIT and MICT revealed that the HIIT intervention was more effective, safer, and better tolerated by patients [31]. Additionally, experiments with HIIT exercise prescriptions of different intensities were conducted, and a progressive individualized model was proposed to be refined and used according to the patient’s specific situation [32]. Several cooperative networks exist to study HIIT in CR, permitting higher levels of communication and cooperation. However, no cooperative relationships were seen between different cooperative groups, suggesting that they focused on internal cooperation.

Among the top 10 most cited articles, the “Cardiovascular risk of high versus moderate-intensity aerobic exercise in coronary heart disease patients” had the highest total number of citations, indicating that this article is of high academic value and reflects research hotspots and trends to a certain extent. This article compares the effects of HIIT and MICT on cardiovascular rehabilitation. The results showed that both HIIT and MICT reduced the incidence of cardiovascular events, but HIIT provided greater cardiorespiratory protection and was more suitable for patients with coronary artery disease [33].

The results showed that Norway, Canada, and the United States were dominant in the field of HIIT in CR. These countries constituted a connection point for exchange and learning between other countries, and this improved academic research and exchange between countries. China only had cooperative relations with the United States, having little exchange with other countries. There was still a big gap between Norway, Canada, and other countries. Future research on HIIT in CR in China could be improved by increased exchanges with researchers from other countries.

The research institute can directly see the leading position and the backbone of the research field of HIIT’s effect on cardiac rehabilitation [24]. In line with previous analyses of authors and countries, the main research institutions were the Norwegian University of Science and Technology, University of Oslo, University of Queensland, and some other universities. These universities are mainly located in countries such as Norway and Canada, further indicating that these regions are among the world leaders in research on the effects of HIIT on cardiac rehabilitation. Most of these research institutions are universities. There were no Chinese universities listed, suggesting a significant gap between China and other countries in this field.

Keywords provide a high-level summary of the research topic and research content and can reveal the core content of an article; they can also reflect both the basic direction of the research content and relationships between individual keywords [34]. Keyword co-occurrence refers to the occurrence of different keywords in the same document and is used to explore hotspots in the research field [35]. The size of the keyword nodes can be observed in the graph, which represents the frequency of keyword occurrence. The higher the keyword frequency in a certain field, the stronger the indication that it is a research hotspot, although not all high frequency keywords have high centrality. Centrality is used to determine the importance of the literature, and higher centrality indicates that the keyword has more influence on the whole field. Those keywords with a centrality greater than 0.1 are considered key nodes and potential research hotspots. One study found that implementing a long-term CR program led to sustained muscle strength gains in men with cardiovascular disease, reducing the decline in health caused by aging and the decline in muscle mass associated with age [36]. Wang Bozhong et al. [37] found that HIIT training after a myocardial infarction in rats could protect cardiomyocytes and blood vessels, reduce inflammation of the heart and peripheral circulation, reduce cardiac fibrosis, and improve cardiac function. Abdelhalem et al. conducted a study of an HIIT model that included aerobic exercise, in order to reduce the prevalence and mortality of patients with coronary heart disease, and were able to increase the aerobic capacity of patients and improve the VO_2max_ of cardiac patients [38]. The main indicators of CR’s effectiveness were exercise capacity, cardiorespiratory fitness, and quality of life. The effectiveness of HIIT was identified via the patient’s physiological response. Studies have shown that HIIT can significantly increase peak oxygen consumption (VO_2peak_) after interventions for coronary artery disease, thereby reducing the risk of death and improving the cardiorespiratory fitness of patients with coronary artery disease. High-intensity interval training can also achieve similar results to MICT in a shorter period of time [39]. VO_2peak_ and maximal oxygen consumption (VO_2max_) represent not only improved circulatory function, but also increased exercise capacity [40,41]. Ellingsen et al. [42] performed a 12-week trial of regular HIIT and showed that it could reverse the left ventricular remodeling of patients with heart failure and improve their aerobic capacity [43]. High-intensity interval training significantly improves cardiac ejection fraction (i.e., increased end-diastolic ventricular volume and decreased end-systolic ventricular volume in patients in heart failure) and can significantly improve systolic-diastolic function [44]. High-intensity interval training is a short and efficient exercise method with a proven safety profile and few adverse effects in heart failure studies [45,46]. After the HIIT intervention, the overall physical health of the patients improved, their physical functions were enhanced, and the long-term intervention produced good results, increasing the patients’ ability to take care of themselves and leading to an improved quality of life. Contraindications to training should be identified and evaluated in advance, permitting the selection of appropriate exercises, strict exercise supervision, and termination of training if necessary [47].

Researchers were more focused on cardiovascular diseases such as coronary artery disease and myocardial infarction, so as to understand the adverse effects of these diseases and the benefits of the combination of drug and exercise therapy. Even when medication is ineffective, HIIT can provide symptom relief, yielding some improvement in exercise capacity and quality of life [48]. Exercise compliance in patients with a cardiovascular disease is a necessary foundation for good long-term recovery, but poor compliance has also been an important problem. The HIIT program can be combined with different exercise modalities to make it easier for patients to stick with the program by adjusting the intensity and enhancing the fun of the exercise, thereby imparting improved body composition, cardiopulmonary function, and glucolipid metabolism. As scholars continue to study HIIT, there has been an increased focus on postoperative rehabilitation tools for patients suffering from coronary artery disease, and myocardial infarction. This shift in research is in line with realistic developmental characteristics [49]. Patients with cardiovascular disease can exhibit symptoms such as autonomic nervous system disorders that lead to an imbalance between sympathetic and naval nerves, resulting in reduced myocardial contractility and increased cardiac load. Implementing HIIT in cardiac rehabilitation programs can improve patients’ cardiovascular health and autonomic nervous system function [50]. Moderate-intensity continuous training, as a traditional form of aerobic exercise, was more commonly applied as endurance training for people, which was beneficial for the rehabilitation of patients with heart disease. Meanwhile, HIIT, in contrast to MICT, can increase aspects such as peak oxygen uptake, heart rate, and cardiac output, all of which contribute to the improvement of the VO_2peak_ and VO_2max_ of patients [51]. Physical activity can improve exercise capacity in people with heart disease. HIIT can be used to reduce the risk factors for cardiovascular disease, such as body fat content and glucolipid metabolism [52]. High-intensity interval training enhances the cardiac function of people with cardiovascular disease, as evidenced by increased cardiac output, beat-to-beat output, and ejection fraction [53]. However, the patient’s physical indicators and physiological responses should be closely monitored during and after HIIT [54]. Exercise intensity should be increased gradually to allow the patient to gradually adapt, ensuring patient safety and avoiding secondary morbidity [55].

The most emergent key term in recent years was “heart transplantation”, indicating that HIIT in heart transplantation will be the latest research hotspot in the field. The burst keywords in recent years were “HIIT” and “oxygen uptake”, indicating that these aspects may also be included among the latest research hotspots in the field. High-intensity interval training had a better impact than “continuous moderate exercise”, improving patients’ VO_2peak_ and aerobic capacity while reducing their cardiovascular mortality. High-intensity interval training programs that are longer in duration and higher in intensity can improve cardiovascular function and increase patients’ tolerance to exercise [56]. High-intensity interval training programs need to be tailored to the individual patient’s abilities and exercise habits in order to maximize compliance and safety while minimizing physiological discomfort [57].

## 5. Conclusions

Our results showed that the number of publications in the field of HIIT in CR has grown rapidly since 2017. This is consistent with the relatively rapid development of the field. The research can be divided into three stages: a stable period from 2004 to 2011, a slow growth period from 2012 to 2016, and a rapid rise from 2017 to the present. Future developments in this research field will be of increasingly high quality and will thereby be able to provide more complete scientific and theoretical data. The leading research institutions in this field are led by the Norwegian University of Science and Technology, the University of Oslo, and the University of Queensland. There is a core group of research institutions in Norway, Canada, and Australia. In journals ranked by number of publications in the field, *Circulation* from the United States was the journal with the highest IF and CiteScore to have published on HIIT in CR. This journal mainly concerns disciplines such as cardiovascular medicine and epidemiology. In our analysis of keyword co-occurrence, clustering time zones and emergent words, it was found that the research theme clusters were mainly focused on “chronotropic response”, “cardiovascular rehabilitation”, and “endothelial function”. Research hotspots have focused on coronary artery disease, exercise capacity, heart failure, cardiorespiratory fitness, and quality of life. High-intensity interval training in heart transplantation may be at the forefront of research in this area, and future studies should focus on this topic. The focus of researchers during each time period was different, indicating that work on HIIT in CR is continuously developing. High-intensity interval training controls cardiovascular disease risk factors—including hypertension, dyslipidemia, and low exercise capacity—and plays a role in improving cardiopulmonary function, ventricular diastolic function, and ventricular remodeling. Patients with cardiovascular disease who engaged in HIIT showed increased VO_2max_ and VO_2peak_, improving their exercise capacity. Patients with impaired cardiac function and restricted daily activities due to their disease also showed significantly improved quality of life following HIIT interventions. Not only did the patients’ self-care ability gradually improve with HIIT, but their family members were also relieved of their care burden.

## 6. Limitations

This paper mainly analyzed the impact of HIIT on CR research from a macro perspective, so as to provide a comprehensive grasp of the research hotspots and trends in this field. CiteSpace was used to analyze the numbers of documents, institutions, countries, authors, and keywords, so there may be a lack of detail in some information, and some small directions are not reflected in the study. The bibliometrics could show the research hotspots in this field, as well as the relationships and contributions be-tween authors, institutions, and countries, to help further explore the development trends in this field. Moreover, CiteSpace can analyze from a global perspective, which is helpful for researchers to systematically grasp the research process and trends in this field. This makes it easier for researchers to make new discoveries on this basis. Finally, the timespan of this study was set from January 2000 to August 2022. The analysis of articles published after August 2022 is lacking, so there will be deviations in the analysis of future research trends.

## Figures and Tables

**Figure 1 ijerph-19-13745-f001:**
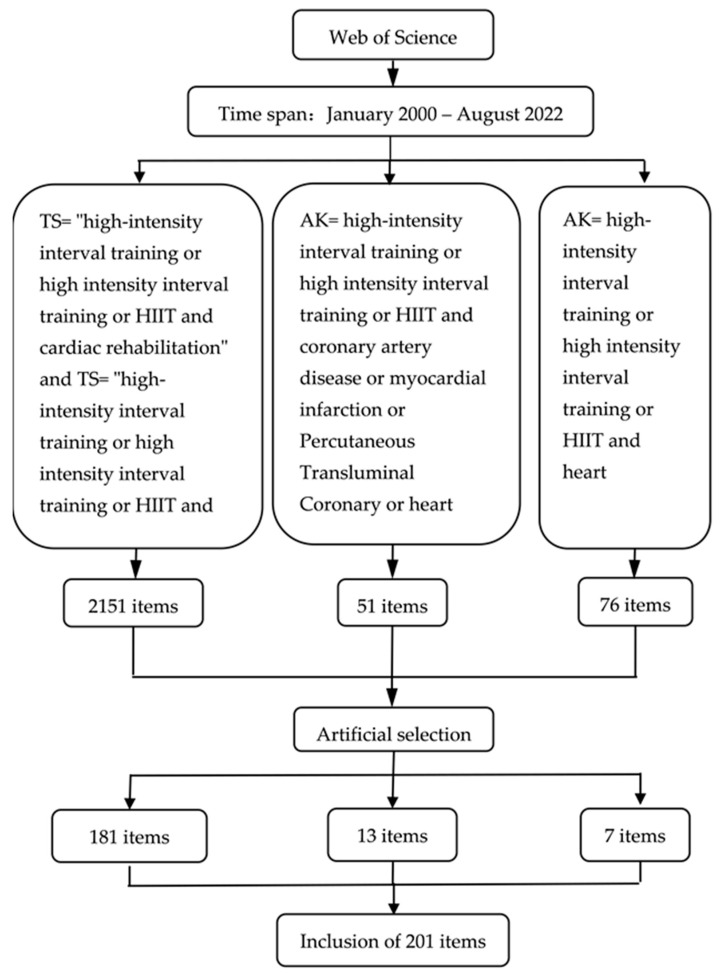
Database search flow chart.

**Figure 2 ijerph-19-13745-f002:**
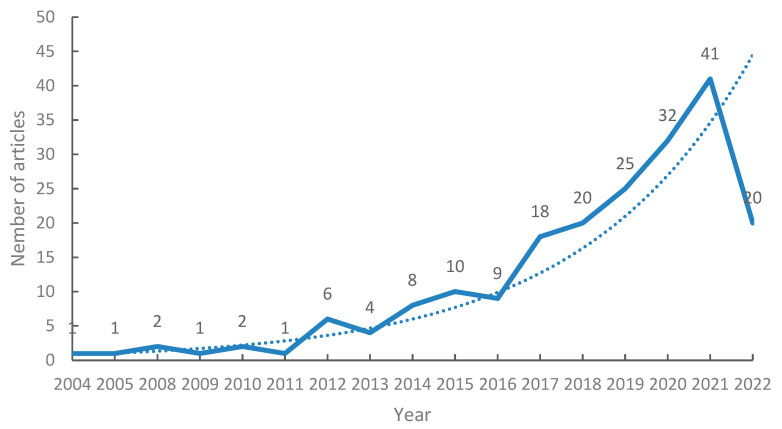
Publication trends over time.

**Figure 3 ijerph-19-13745-f003:**
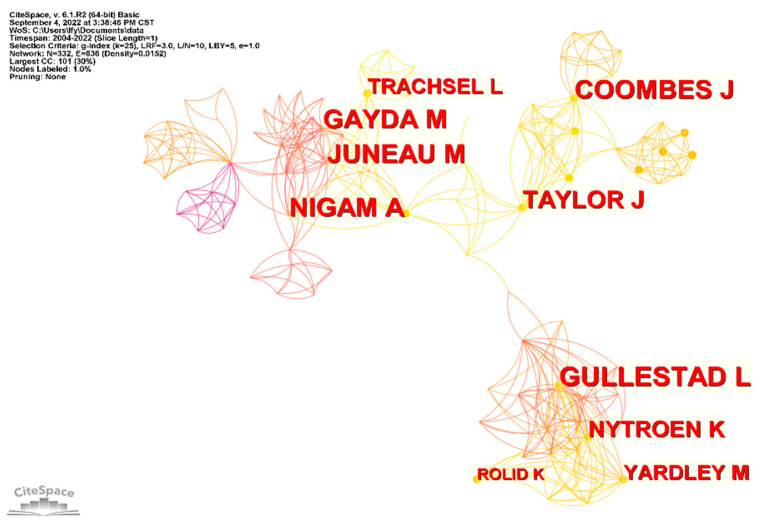
Author collaboration chart.

**Figure 4 ijerph-19-13745-f004:**
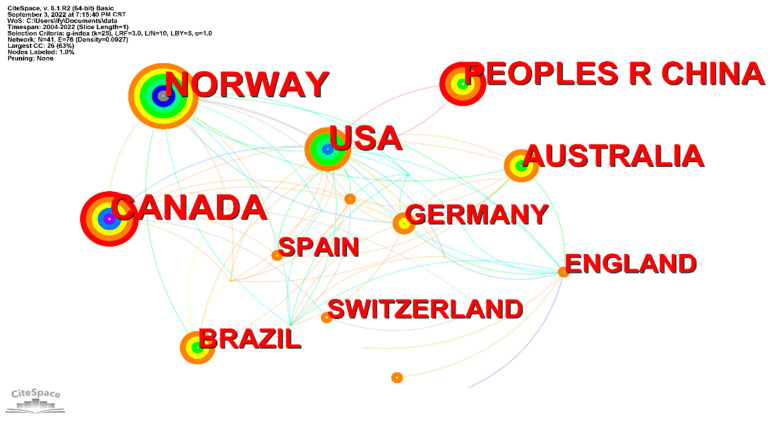
Countries collaboration chart.

**Figure 5 ijerph-19-13745-f005:**
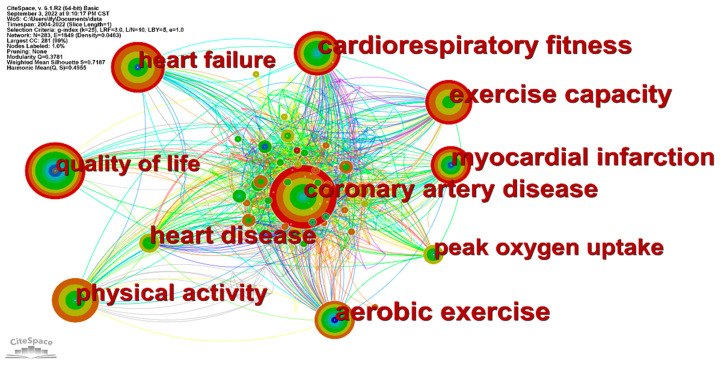
Keyword co-occurrence knowledge map.

**Figure 6 ijerph-19-13745-f006:**
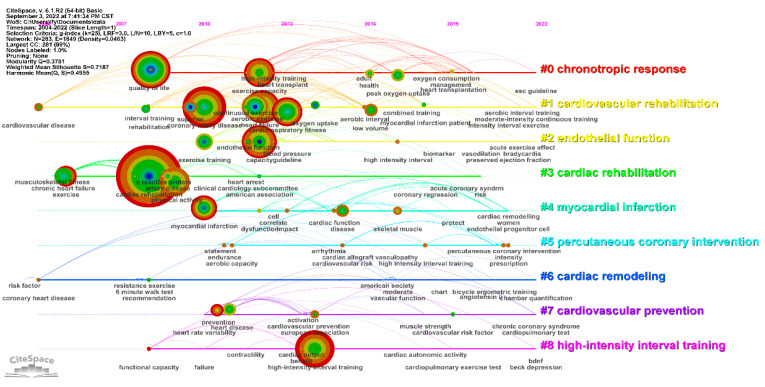
Keyword clustering and visual time-zone mapping.

**Figure 7 ijerph-19-13745-f007:**
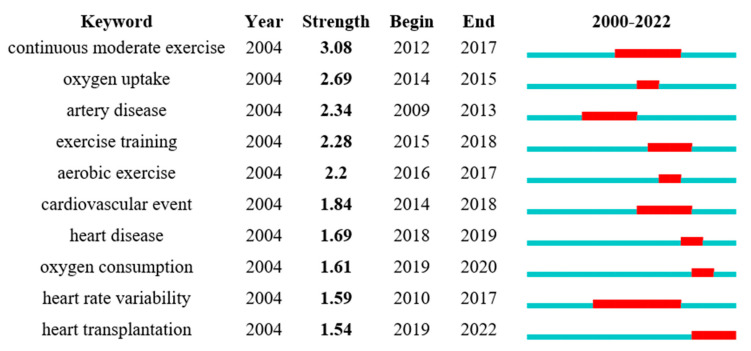
Top 10 keyword bursts.

**Table 1 ijerph-19-13745-t001:** Top 9 journals by publications.

Rank	Journal	Publications	Citations	Country	Category Zone	IF (2021)	CiteScore
1	*European Journal of Preventive Cardiology*	15	308	United Kingdom	Q1	8.526	9.8
2	*International Journal of Cardiology*	9	184	Ireland	Q2	4.039	7.0
3	*Journal of Cardiopulmonary Rehabilitation and Prevention*	9	164	United States	Q2	3.646	3.9
4	*Frontiers in Cardiovascular Medicine*	6	6	Switzerland	Q2	5.846	8.2
5	*Annals of Physical and Rehabilitation Medicine*	5	142	France	Q1	5.393	5.7
6	*Applied Physiology Nutrition and Metabolism*	5	41	Canada	Q2	3.016	4.5
7	*Circulation*	5	505	United States	Q1	39.918	40.5
8	*Medicine*	4	3	United States	Q3	1.817	2.7
9	*ESC Heart Failure*	4	19	United States	Q2	3.612	3.8

**Table 2 ijerph-19-13745-t002:** Top 10 authors by publications.

Rank	Author	Publications	Country	Total Citations	Average Citations
1	Gullestad, L	11	Norway	199	18.1
2	Nytroen, K	11	Norway	199	18.1
3	Juneau, M	11	Canada	426	38.7
4	Nigam, A	11	Canada	426	38.7
5	Gayda, M	10	Canada	254	25.4
6	Rolid, K	8	Norway	122	15.3
7	Coombes, J	8	Australia	158	19.8
8	Taylor, J	8	United States	59	7.4
9	Trachsel, L	7	Switzerland	28	4.0
10	Yardley, M	7	Norway	83	11.9

**Table 3 ijerph-19-13745-t003:** Top 10 papers with the most citations.

Rank	Title	Author	Periodicals	Frequency of Citations	Year
1	Cardiovascular risk of high- versus moderate-intensity aerobic exercise in coronary heart disease patients	Rognmo, O	*Circulation*	269	2012
2	High-intensity interval training in patients with heart failure with reduced ejection fraction	Ellingsen, O	*Circulation*	195	2017
3	Effectiveness of high-intensity interval training for the rehabilitation of patients with coronary artery disease	Warburton, DER	*American Journal of Cardiology*	174	2005
4	High-intensity interval training in cardiac rehabilitation	Guiraud, T	*Sports Medicine*	172	2012
5	High intensity interval training for maximizing health outcomes	Karlsen, T	*Progress in Cardiovascular Diseases*	125	2017
6	High-intensity interval training vs. moderate-intensity continuous exercise training in heart failure with preserved ejection fraction: a pilot study	Angadi, SS	*Journal of Applied Physiology*	111	2015
7	Low-volume, high-intensity interval training in patients with CAD	Currie, KD	*Medicine and Science in Sports and Exercise*	110	2013
8	Interval training versus continuous exercise in patients with coronary artery disease: a meta-analysis	Elliott, AD	*Heart Lung and Circulation*	108	2015
9	High-intensity interval training versus moderate intensity continuous training within cardiac rehabilitation: a systematic review and meta-analysis	Hannan, AL	*Open Access Journal of Sports Medicine*	103	2018
10	High-intensity interval training may reduce in-stent restenosis following percutaneous coronary intervention with stent implantation: a randomized controlled trial	Munk, PS	*American Heart Journal*	103	2009

**Table 4 ijerph-19-13745-t004:** Top 10 countries by publications.

Rank	Country	Number of Publications
1	Norway	33
2	Canada	30
3	United States	29
4	China	21
5	Australia	19
6	Germany	13
7	Brazil	13
8	United Kingdom	12
9	Spain	12
10	Switzerland	11

**Table 5 ijerph-19-13745-t005:** Top 10 institutions in terms of number of articles issued.

Rank	Country	Institutions	Number of Publications
1	Norway	Norwegian University of Science and Technology	14
2	Norway	University of Oslo	13
3	Australia	University of Queensland	12
4	Canada	University of Montreal	10
5	United States	Mayo Clinic	8
6	Canada	Montreal Heart Institute	7
7	Canada	McMaster University	6
8	Norway	Oslo University Hospital	6
9	Norway	Norway Health Association	5
10	Australia	Sunshine Coast University Hospital	5
10	Canada	Hamilton Health Science	5
10	Germany	Technical University of Munich	5

**Table 6 ijerph-19-13745-t006:** Top 10 keywords co-occurrence frequency.

Rank	Keywords	Number of Occurrences	Centrality	Year of Occurrence
1	Coronary artery disease	59	0.07	2010
2	Exercise capacity	42	0.13	2012
3	Heart failure	42	0.05	2012
4	Cardiorespiratory fitness	38	0.08	2013
5	Physical activity	37	0.06	2009
6	Quality of life	37	0.05	2008
7	Aerobic exercise	36	0.14	2012
8	Myocardial infarction	31	0.10	2010
9	Heart disease	19	0.08	2011
10	Peak oxygen uptake	18	0.07	2017

**Table 7 ijerph-19-13745-t007:** List of keyword clusters.

Rank	Size	Silhouette	Cluster	Keywords (Partial)
#0	44	0.602	Chronotropic response	Heart transplantation; oxygen consumption; heart-related quality; myocardial infarction
#1	41	0.665	Cardiovascular rehabilitation	Cardiovascular disease; coronary artery disease; skeletal muscle power; myocardial infarction
#2	41	0.695	Endothelial function	Acute coronary syndrome; energy expenditure; chronic heart failure; cardiopulmonary exercise test; intermittent exercise
#3	36	0.895	Cardiac rehabilitation	Exercise prescription; cardiovascular disease; peak oxygen uptake; cardiorespiratory fitness
#4	30	0.772	Myocardial infarction	Dysfunction; cell; rehabilitation; high-intensity interval training; aerobic exercise
#5	26	0.705	Percutaneous coronary intervention	Coronary artery disease; cardiac rehabilitation; clinical outcomes
#6	23	0.686	Cardiac remodeling	Ergometric training; coronary artery bypass; chronic heart failure
#7	21	0.813	Cardiovascular prevention	Exercise prescription; inspiratory muscle training; moderate-intensity aerobic continuous exercise training; coronary artery disease
#8	19	0.775	High-intensity interval training	Health-related quality; myocardial infarction; maximal oxygen consumption; peak oxygen consumption

## Data Availability

Not applicable.

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
