# Peer review of "A Bibliometric Analysis of High-Intensity Interval Training in Cardiac Rehabilitation"

_ijerph, 2022, doi:10.3390/ijerph192113745_

Round 1
Reviewer 1 Report
Li et. al, have presented the manuscript in an understandable manner and they have provided enough evidence for their claims. However, there are some questions that exist. First of all, this is a bibliometric analysis, and we cannot infer precisely any outcome of the study. The authors have not mentioned in a particular manner how the sample population was considered for the study.
1. Have they considered any particular age group, weight, race, and gender for sampling?
2. Please mention the limitation of this manuscript.
3. Have the authors noticed any keywords related to the cause of cardiovascular problems in their analysis?
Author Response
Revision Notes
Dear Reviewer,
Thank you for your comments concerning our manuscript. We are so grateful for the rigorious reviewer. Those comments are all valuable and very helpful for revising and improving our paper, as well as the important guiding significance of our research. We tried our best to improve the manuscript and made some changes to the manuscript, and hoped that the corrections would meet with approval.
The manuscript has undergone English language editing. The modified documents have tracked changes to highlight the revisions. The main corrections in the paper and the responds to the reviewer’s comments are as follows.
Response to Reviewer 1 Comments
Reviewer Comments for the Author...
Li et. al, have presented the manuscript in an understandable manner and they have provided enough evidence for their claims. However, there are some questions that exist. First of all, this is a bibliometric analysis, and we cannot infer precisely any outcome of the study. The authors have not mentioned in a particular manner how the sample population was considered for the study.
- Have they considered any particular age group, weight, race, and gender for sampling?
Authors’ response:
We thank the reviewer for your comments. We did not consider the sample population of specific age groups, weight, race, or gender. This study is an overall grasp of the hot spots and trends in the field of high-intensity interval training in cardiac rehabilitation, so some information about this field is not explained in detail. We analyzed the volume of publications, institutions, countries, authors, and keywords in the field with the help of CiteSpace software. It has been explained in the added research limitations of this paper. This study will not only help researchers extract potentially valuable information for in-depth investigation, but also provide meaningful guidance for their selection of cutting-edge topics. For researchers, it is helpful to dig out the hot topics and cutting-edge topics from the vast literature and then conduct research activities in conjunction with the needs and existing research.
- The abstract lacks the objective of the study. It should be made explicit.
Authors’ response:
Thanks to the reviewer for raising this important issue. We would like to thank the reviewer for his rigor. This paper mainly analyzed the impact of HIIT on CR research from a macro perspective, so as to provide a comprehensive grasp of the research hotspots and trends in this field. CiteSpace was used to analyze the numbers of documents, institutions, countries, authors, and keywords, so there may be a lack of detail in some information, and some small directions are not reflected in the study. The bibliometrics could show the research hotspots in this field, as well as the relationships and contributions between authors, institutions, and countries, to help further explore the development trends in this field. Moreover, CiteSpace can analyze from a global perspective, which is helpful for researchers to systematically grasp the research process and trends in this field. This makes it easier for researchers to make new discoveries on this basis. Finally, the timespan of this study was set from January 2000 to August 2022. The analysis of articles published after August 2022 is lacking, so there will be deviations in the analysis of future research trends. These limitations have been added to the article.
Please see page 15, line 400-412.
- Have the authors noticed any keywords related to the cause of cardiovascular problems in their analysis?
Authors’ response:
We would like to thank the reviewer for his rigor. When analyzing the keywords in this study, we ranked the keywords according to their frequency. The top ten keywords in terms of frequency of occurrence were then organized into tables and images and analyzed. Because of the low frequency of keywords related to cardiovascular problems, they did not appear in the top 10. Therefore, the keywords of the cause of cardiovascular problems were not focused on during the analysis.

Reviewer 2 Report
This manuscript describes a bibliometric analysis of the use of high-intensity interval training in cardiac rehabilitation.
The first sentence in the Abstract must be rewritten due to poor grammar.
Line 37, 41, 46, 56, 59, 295, 303, 326, 328, 343, 345, 365, 368: Generally it acronyms should not be used at the start of a sentence.
Line 41-42: Poor grammar, please rewrite.
Line 43: Change to "...aims to improve daily living and quality of life..."
Line 75: Change to "Data from relevant literature used in this study were extracted from the Web of Science (WOS) database,..."
Lines 78-79: This sentence is confusing, "..were included." Do you mean "...were excluded."?
Lines 79 and 81: What is TS?
Line 83: Change to "The second step was to search author keywords (AK)..."
Line 87: Change to "The third step was to run an author keyword search..."
Lines 89-90: The meaning of TS should have been in brackets the first time it was used and the AK has already been stated so no need for it here.
All figures and tables are clear and helpful for the reader.
Discussion section is well written and conclusions are based on findings of the combined research in the use of HIIT for CR.
This manuscript provides useful information from the literature on the current body of knowledge regarding the safe use of HIIT for CR.
Author Response
Revision Notes
Dear Reviewer,
Thank you for the reviewer’ comments concerning our manuscript. Those comments are all valuable and very helpful for revising and improving our paper.
It is my pleasure to meet the reviewers who are meticulous. It has a great guiding role in our future scientific research and writing papers, which has benefited us a lot. We appreciate for Reviewers’ warm work earnestly, and hope that the correction will meet with approval.
The manuscript has undergone English language editing. The modified documents have tracked changes to highlight the revisions. The main corrections in the paper and the responds to the reviewer’s comments are as follows.
Response to Reviewer 2 Comments
Reviewer Comments for the Author...
This manuscript describes a bibliometric analysis of the use of high-intensity interval training in cardiac rehabilitation.
- The first sentence in the Abstract must be rewritten due to poor grammar.
Authors’ response:
We thanked the reviewer for the proposal, and based on your suggestion, we have changed this sentence.
Please see page 1, line 12-13.
- Line 37, 41, 46, 56, 59, 295, 303, 326, 328, 343, 345, 365, 368: Generally it acronyms should not be used at the start of a sentence.
Authors’ response:
We would like to thank the reviewer for his rigor. We have changed the acronym at the beginning of the sentence to the full name as per your request.
Please see page 1, line 35-36, line38, line43; page 2, line49, line59, line63-64; page 13, line312, line318, line 321, line346; page 14, line353-354, line363-364, line366, line368, line388, line391.
- Line 41-42: Poor grammar, please rewrite.
Authors’ response:
We would like to thank the reviewer for his rigor. We have made changes to the syntax of the sentence.
Please see page 1, line 43-44.
- Line 43: Change to "...aims to improve daily living and quality of life...".
Authors’ response:
Thank you very much. We have made the changes as you requested.
Please see page 2, line 45-46.
- Line 75: Change to "Data from relevant literature used in this study were extracted from the Web of Science (WOS) database..."
Authors’ response:
This is an excellent suggestion. According to your opinion. we have changed this sentence.
Please see page 2, line 80-81.
- Lines 78-79: This sentence is confusing, “...were included." Do you mean "...were excluded."?
Authors’ response:
Thank you very much. The meaning we want to express is “... were excluded". This was the exclusion criteria we used when conducting literature screening. We have made changes to address this issue.
Please see page 2, line 83-85.
- Lines 79 and 81: What is TS?
Authors’ response:
Thank the reviewer for raising this question. TS refers to the topic, which we have already added in the text.
Please see page 2, line 86.
- Line 83: Change to "The second step was to search author keywords (AK)..."
Authors’ response:
Thanks to the reviewer. We have changed this sentence.
Please see page 2, line 89-90.
- Line 87: Change to "The third step was to run an author keyword search..."
Authors’ response:
Thank the reviewer for your advice. We changed it.
Please see page 2, line 94.
- Lines 89-90: The meaning of TS should have been in brackets the first time it was used and the AK has already been stated so no need for it here.
Authors’ response:
Thank the reviewer for carefully pointing out the problems in the article. We have completed the meaning of TS in the previous section, so we have deleted this sentence here.
Please see page 2, line 96-97.

Round 2
Reviewer 1 Report
The changes have been made by authors are satisfactory.